# Changes in inflammatory and vasoactive mediator profiles during valvular surgery with or without infective endocarditis: A case control pilot study

Mahmoud Diab[1,2], Raphael Tasar[1], Christoph Sponholz[3], Thomas Lehmann[4], Mathias W. Pletz[2,5], Michael Bauer[2,3], Frank M. Brunkhorst[2,4], Torsten Doenst[1]*

**1** Department of Cardiothoracic Surgery, Jena, Germany, **2** Center for Sepsis Control and Care, Jena, Germany, **3** Department of Anaesthesiology and Critical Care Medicine, Jena, Germany, **4** Center of Clinical Studies, Jena, Germany, **5** Institute for Infectious Diseases and Infection Control, Jena University Hospital– Friedrich Schiller University of Jena, Jena, Germany

* doenst@med.uni-jena.de

**Data Availability Statement:** Some of patient´s data cannot be shared publicly because of the patient´s data protection policy of our institution.

## Abstract

### Background

More than 50% of patients with infective endocarditis (IE) develop an indication for surgery. Despite its benefit, surgery is associated with a high incidence of multiple organ dysfunction syndrome (MODS) and mortality, which may be linked to increased release of inflammatory mediators during cardiopulmonary bypass (CPB). We therefore assessed plasma cytokine profiles in patients undergoing valve surgery with or without IE.

### Methods

We performed a prospective case-control pilot study comparing patients undergoing cardiac valve surgery with or without IE. Plasma profiles of inflammatory mediators were measured at 7 defined time points and reported as median (interquartile). The degree of MODS was measured using sequential organ failure assessment (SOFA) score.

### Results

Between May and December 2016 we included 40 patients (20 in each group). Both groups showed similar distribution of age and gender. Patients with IE had higher preoperative SOFA (6.9± 2.6 vs 3.8 ± 1.1, p<0.001) and operative risk scores (EuroSCORE II 18.6±17.4 vs. 1.8±1.3, p<0.001). In-hospital mortality was higher in IE patients (35% vs. 5%; p<0.001). Multiple organ failure was the cause of death in all non-survivors. At the end of CPB, median levels of following inflammatory mediators were higher in IE compared to control group: IL-6 (119.73 (226.49) vs. 24.48 (40.09) pg/ml, $p = 0.001$); IL-18 (104.82 (105.99) vs. 57.30 (49.53) pg/ml, $p<0.001$); Mid-regional pro-adrenomedullin (MR-proADM) (2.06 (1.58) vs. 1.11 (0.53) nmol/L, $p = 0.003$); MR- pro-atrial natriuretic peptide (MR-proANP) (479.49 (224.74) vs. 266.55 (308.26) pmol/l, $p = 0.028$). IL-1β and TNF- α were only detectable in IE patients and first after starting CPB. Plasma levels of IL-6, IL-18, MRproADM, and MRproANP during CPB were significantly lower in survivors than in those who died.

However, all research data will be available to be used by the Center for Sepsis Control and Care (CSCC) and its international partners. In addition, study protocol, anonymized demographic data of the patients, and statistical analysis can be provided to researchers outside CSCC, who provide a study protocol which is approved by the Data Steering Committee (DSC), located at the Center for Clinical Studies (CCS), Jena University Hospital (https://www.uniklinikum-jena.de/zks/en/). Members of the DSC consist of the responsible data manager, biostatical, project manager and the scientific director of the CCS This data is sufficient to replicate the study. Requests should be sent to the head of the DSC: frank.brunkhorst@med.uni-jena.de. Data from this study will be archived for 10 years on the servers of the Center for Clinical Studies, Jena University Hospital to be available for further use by other investigators, as long as there is no conflict with the copyrights of the publisher. After 10 years the data will be archived; however can still be accessed after approval from the aforementioned DSC.

**Funding:** Laboratory analyses were funded by B.R. A.H.M.S. GmbH, part of Thermo Fischer Scientific. In addition, B.R.A.H.M.S. GmbH took over the costs for data and project management in the Center of clinical studies, University hospital of Jena

**Competing interests:** All other authors declare no competing interests

## Conclusion

The presence of infective endocarditis during cardiac valve surgery is associated with increased inflammatory response as evident by higher plasma cytokine levels and other inflammatory mediators. Actively reducing inflammatory response appears to be a plausible therapeutic concept.

## Trial registration

ClinicalTrials.gov, ID: NCT02727413.

## Introduction

Infective endocarditis (IE) affects 1-10/100,000 persons per year and is associated with up to 40% in-hospital mortality [1–3]. Surgical treatment is necessary in about 50% of patients and is associated with in-hospital mortality as high as 15–25% and 1-year mortality of 40% [1, 4]. The postoperative course of patients with IE is often complicated with a varying degree of circulatory failure i.e. hypotension, decreased systemic vascular resistance, despite high cardiac output, adequate fluid resuscitation, and adrenergic vasopressor administration which can progress to septic shock in up to 10–28% of cases [5–7].

Cardiopulmonary bypass (CPB) is an essential part of cardiac surgery for IE and has been shown to cause a systemic inflammatory response which may result in severe organ dysfunction and increased postoperative mortality *[19]* [8]. In addition to CPB, contact of blood with the operative bed induce inflammatory response and the release of cytokines [9]. Because the operative bed is infected in IE, this may induce a stronger inflammatory reaction and subsequent cytokine release than in non-infected. Unfortunately, available information on profiles of cytokines in IE is scarce. [10, 11], and, according to our knowledge, there are no data available that compare peri-operative cytokines profile between patients undergoing cardiac surgery for IE and non-infectious valve disease.

Such data may provide valuable knowledge and may assist to develop measures aiming at perioperative reduction of cytokines and, thereby, may improve survival of endocarditis patients undergoing cardiac surgery.

## Methods

### Study design and patients

We performed a case-control observational prospective pilot study on patients undergoing cardiac valvular surgery for either definite infective endocarditis according to the modified Duke criteria (20 patients)[12], or VHD without endocarditis. Exclusion criteria were glucocorticoid or any other immunosuppressive therapy, severe neutropenia <1000/mm$^3$, patients younger than 18 years-old, or pregnancy. Fig 1 shows the flowchart of the study.

### Ethics approval

The study was approved by the ethics committee of the Jena University Hospital, Germany (reference number 4700-02/16).

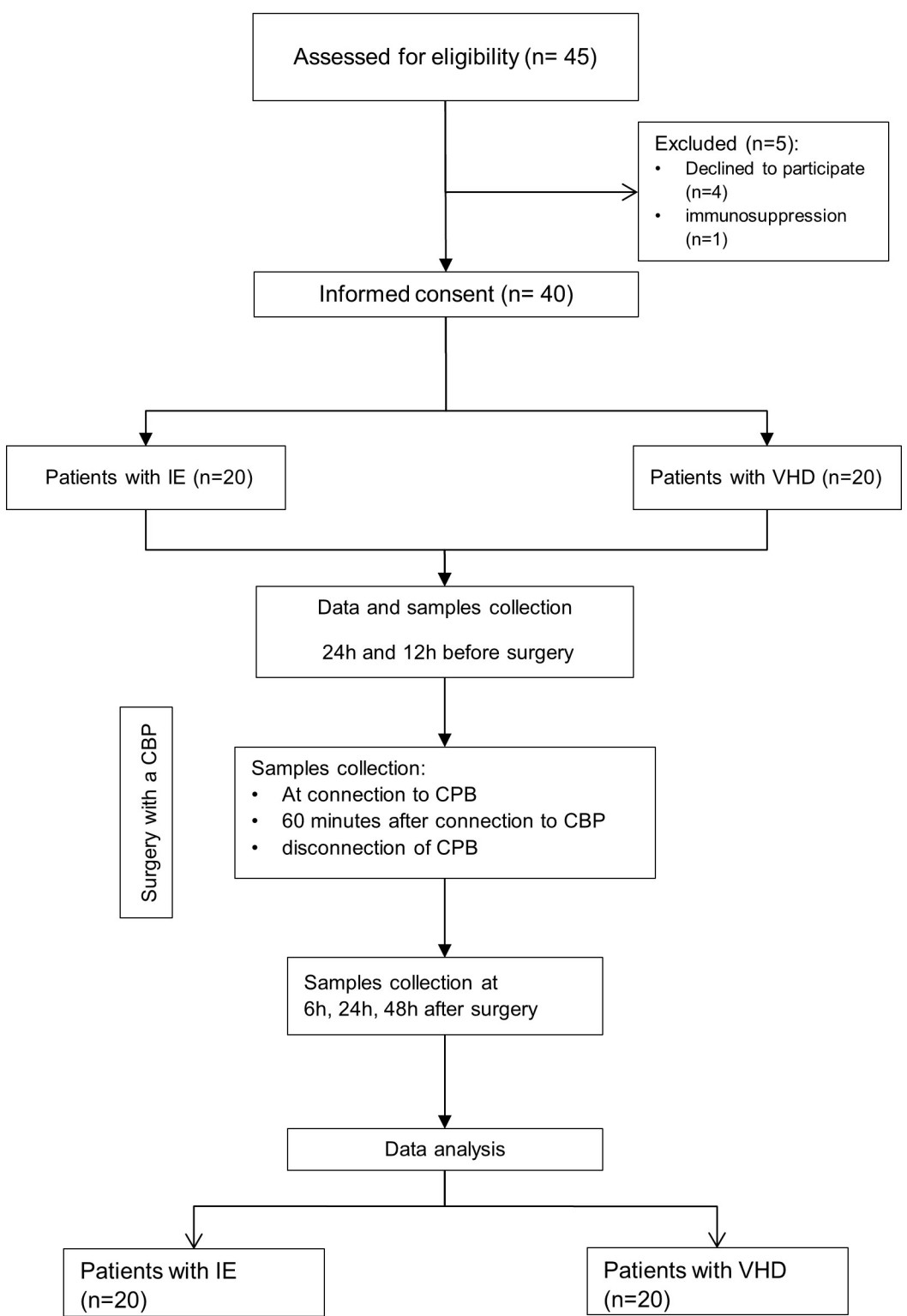

**Fig 1. IE: Infective endocarditis; VHD: Valvular heart disease.**

## Obtaining the informed consent

A written informed consent was obtained from each patient before inclusion in the study according to § 28 of the Declaration of Helsinki.

## Endpoints

1. Plasma profiles of inflammatory biomarkers: Procalcitonin, C-reactive protein, C-terminal proendothelin-1(CT-peoET-1), tumor necrosis factor alpha (TNFα), interleukin (IL)-1β, IL-6, IL-10, IL-18.

2. Plasma profiles of inflammation-related vasoactive mediators: midregional pro-Adrenome-dullin (MR-proADM), copeptin pro-Arginine Vasopressin (CT-proAVP), midregional pro-Atrial natriuretic Peptide (MR-proANP).

3. Changes in organ dysfunction during the 1st and 2nd post-operative days, disclosed by Δ SOFA score as compared to pre-surgery status.

4. Use and duration of renal replacement therapy.

5. Cumulative doses of concomitant medications (vasopressors, corticoids, prostaglandins) applied during the surgery and over 48 h thereafter.

6. In-hospital mortality within 30 days post-surgery.

## Data and samples collection

Pre-operative data including assessment of surgical risk (EuroScore), pre-surgical co-morbidity (Charlson score) and acute organ dysfunction (SOFA score), operative data, and postoperative data including SOFA score, cumulative doses of concomitant medications, and use of renal replacement therapy were recorded in a computerized clinical research form (eCRF). Data acquisition was done via web application into the study management software OpenClinica®.

Samples were collected at the following time points:

- 12 to 24 hours before transfer to the operating theatre

- At connection to the CPB

- 60 minutes after connection to the CPB,

- disconnection of the CPB

- 6, 24 and 48 hours after the end of the operation

All proteins were measured in EDTA plasma. The biomarkers MR-proADM (nmol/L), MR-proANP (pmol/L), CT-proET-1 (pmol/L), CRP (µg/mL), PCT (µg/L) and CT-proAVP (pmol/L) were measured using the B·R·A·H·M·S™ MR-proADM KRYPTOR™, B·R·A·H·M·S™ MR-proANP KRYPTOR™, B·R·A·H·M·S™, CT-proET-1 KRYPTOR™, B·R·A·H·M·S™ CRPus KRYPTOR™, B·R·A·H·M·S™ PCT sensitive KRYPTOR™ and B·R·A·H·M·S™ CT- proAVP KRYPTOR™ (B·R·A·H·M·S GmbH, part of Thermo Fisher Scientific), respectively. The biomarkers IL-6, IL-1ß, IL-10, IL-18 and TNF-α (pg/mL) were measured using the IL-6 Human ProcartaPlex™ Simplex Kit, IL-1ß Human ProcartaPlex™ Simplex Kit, IL-10 Human Procarta-Plex™ Simplex Kit, IL-18 Human ProcartaPlex™ Simplex Kit, TNF-α Human ProcartaPlex™ Simplex Kit (Bender MedSystems GmbH, part of Thermo Fisher Scientific), in multiplex

Information on in-hospital mortality over 30 days was gathered from patient records or, in case of transfer to other facilities, through confidential inquiries conducted by the Principal Investigator.

Data processing employed the software OpenClinica, which fulfils the regulatory requirements (GCP, 21 CFR Part 11). Each subject was given an unambiguous patient identification number to ensure pseudonymized data analysis

Information on in-hospital mortality over 30 days was gathered from patient records or, in case of transfer to other facilities, through confidential inquiries conducted by the Principal Investigator.

Data processing employed the software OpenClinica, which fulfils the regulatory requirements (GCP, 21 CFR Part 11). Each subject was given an unambiguous patient identification number to ensure pseudonymized data analysis.

Since there are no data from other trials for this patient population available a generic approach was used to estimate the required sample size. For the non-parametric sample size planning the effect size is given as relative treatment effect p [13]. The null hypothesis of the statistical test is $H_0$: p = 0.5 and the two-sided alternative is $H_1$: p ≠ 0.5. For a sample size of 2 x 20 patients, our study has a power of ≥ 80% to detect p ≥ 0.75 (or p ≤ 0.25) at a two-sided significance level of 5% for the non-parametric test (Mann-Whitney U test, software: nQuery Advisor 7.0).

## Statistical analysis

Baseline patient characteristics were reported as mean + standard deviation or median (25th - 75th percentile) for continuous variables. For categorical variables, data were reported as frequencies and percentages. Mann-Whitney U tests were performed to compare continuous variables between two groups. Fisher's exact test was used to compare in-hospital mortality rates of endocarditis and VHD patients. Spearman's rank correlation was used to assess the association of cytokines and vasopressors. All analyses are exploratory and no adjustment for multiplicity was applied, the level of significance was set at 5% for each test. All statistical analyses were done using SAS 9.3 (SAS Institute, Cary NC)).

## Results

Between June and December 2016 we prospectively included 40 patients who underwent valvular surgery either for infective endocarditis (n = 20) or non-infectious valvular heart disease (n = 20).

Table 1 shows pre-operative patient characteristics of the study population divided into patients with IE and those with non-infectious VHD. Both groups were similar in the distribution of age and gender. However, patients in the IE group had higher operative risk (Euro-SCORE II 18.6±17.4 vs. 1.8±1.3, $p$ <0.001), higher Charlson comorbidity index (5.75±3.46vs. 3.65±1.98, $p$ = 0.039), and higher SOFA score (7 (IQR 5–7) vs. 4 (IQR 3–4.5), p< 0.001).

Table 2 shows operative procedures as well as outcome of the study population divided into patients with IE and those with non-infectious VHD. Operative procedures were similar in both groups. The mean duration of CPB was longer in IE group compared to the control group (118.7± 57 minutes vs. 99.6 ± 35.2 minutes, $p$ = 0.002). More than half of patients in VHD were operated using minimally invasive approach compared to only 15% in the IE group. The mean length of stay in the intensive care unit was longer in the IE group, however, the difference was not statistically significant (10.6 ± 9.0 days vs. 4.7±2.9 days, p = 0.052). The in-hospital mortality was significantly higher in IE group (35% versus 5%, p = 0.044).

**Table 1. Pre-operative patients' characteristics.**

| | Endocarditis N = 20 | VHD N = 20 | P |
|---|---|---|---|
| Age (yrs) | 63.6±9.5 | 66.5±10.3 | 0.464 |
| Sex (m/f) | 13/7 | 13/7 | 1.000 |
| BMI (kg/m2) | 27.3±5.2 | 27.1±5.5 | 0.482 |
| EuroSCORE II | 18.6±17.4 | 1.8±1.3 | < 0.001 |
| Charlson Morbidity Index | 5.75±3.46 | 3.65±1.98 | 0.039 |
| Neurological disorders | 6 (30%) | 1 (5%) | 0.091 |
| SOFA score median (IQR) | 7 (5–7) | 4 (3–4.5) | < 0.001 |
| Diabetes mellitus | 17 (85) | 16 (80) | 1.000 |
| Hypertension | 16 (80) | 17 (85) | 1.000 |
| COPD | 4 (20) | 2 (10) | 0.661 |
| PAVD | 4 (20) | 2 (10) | 0.661 |
| Myocardial infarct | 1 (5) | 2 (10) | 1.000 |
| Left ventricular ejection fraction (%) | 62.2±8.7 | 56.6±12.8 | 0.206 |
| Pulmonary artery pressure (mmHg) | 27.0±2.2 | 32.7±11 | 0.172 |
| NYHA III/IV | 11 (55%) | 7 (35%) | 0.341 |
| Type of admission | | | <0.001 |
| • Elective | 0 | 20 (100.0) | |
| • Urgent | 15 (75.0) | 0 | |
| • emergency | 5 (25.0) | 0 | |
| Poor mobility | 10 (50.0) | 1 (5.0) | 0.003 |
| Critical Preoperative State | 12 (60.0) | 1 (5.0) | <0.001 |
| Anti-coagulation | 5 (25%) | 5 (25%) | 1.000 |
| Antiplatelet | 6 (30%) | 5 (25%) | 1.000 |
| Hemodialysis | 2 (10.0) | 1 (5.0) | 1.000 |
| Re-operation | 6 (30.0) | 1 (5.0) | 0.091 |
| Coronary artery disease | 1 (5.0) | 0 | 1.000 |
| Administration of | | | |
| • Norepinephrine | 2 (10.0) | 0 | 0.487 |
| • Epinephrine | 0 | 0 | 1.000 |
| • Vasopressin | 1 (5.0) | 0 | 1.000 |

Data are presented as mean± Standard deviation (SD) or n (%); VHD: valvular heart disease; BMI: body mass index; EuroSCORE: European System for Cardiac Operative Risk Evaluation; SOFA: sequential organ failure assessment; IQR: interquartile; COPD: chronic obstructive pulmonary disease; PAVD: peripheral arterial vascular disease; NYHA: New York Heart Association

Fig 2A shows SOFA scores of endocarditis and VHD patients within 24h pre-operative, on the 1st post-operative day, and on the 2nd post-operative day. Post-operative SOFA score was significantly higher in endocarditis patients compared to the control group (1st post-operative day: 12 (9–13) vs. 6.5 (5.5–8.5), p<0.001; 2nd postoperative day: 10.5 (6–12) vs. 6 (6–7), p = 0.014). Changes between postoperative and pre-operative SOFA scores (ΔSOFA) within the same group were non-significantly different in both groups on the 1st post-operative day (5 (3–10) in the IE vs. 3 (3.5–10) in VHD, p = 0.173) as well as on the 2nd post-operative (3 (3–12) in IE vs. 3 (1.5–8) in VHD, p = 0.577).

Fig 2B shows SOFA subscores within 24 h pre-operative in IE patients compared to control group. The liver was the most common organ affected in patients with IE (84% of IE patients had hepatic failure compared to 16% in VHD patients). Respiratory failure was the 2nd most common organ failure in patients with IE (40%), followed by cardiovascular failure (25%).

**Table 2. Operative data and outcome.**

| | Infective endocarditis N = 20 | Valvular heart disease N = 20 | P |
|---|---|---|---|
| Valvular surgery | | | |
| Mitral valve | 13 (65) | 7 (35) | 0.113 |
| Aortic valve | 10 (50) | 12 (60) | 0.751 |
| Tricuspid | 5 (25) | 5 (25) | 1.000 |
| Concomitant surgery | | | 1.000 |
| CABG | 1(5) | 1 (5) | |
| Replacement of ascending aorta | 1(5) | 0 | |
| Minimally invasive approach | 3 (15) | 11 (55) | 0.019 |
| Bypass-time (min) | 118.7±57 | 99.6±35.2 | 0.002 |
| Duration of X calmp | 62.10±35.96 | 64.80±24.65 | 0.783 |
| ICU-length of stay (days) | 10.6±9.0 | 4.7±3.0 | 0.052 |
| In-hospital mortality | 7 (35%) | 1(5%) | 0.044 |
| Post-op. Hemodialysis | 1(5) | 1 (5) | 1.000 |

Data are presented as mean± Standard deviation (SD) or n (%); ICU: intensive care unit; X clamp: aortic cross clamp; CABG: coronary artery bypass grafting

Fig 3 shows the cumulative doses of norepinephrine administered during the 24 hours pre-operative (-24) as well as during the 1st post-operative day (24) and the 2nd post-operative (48) days in patients with IE compared to VHD patients.

There was no difference pre-operatively in norepinephrine doses between the two groups. The cumulative dose of norepinephrine during the 1$^{st}$ post-operative day in IE patients was substantially higher in IE patients compared to patients with VHD; however, the difference was not statistically significant. During the 2$^{nd}$ postoperative day, the cumulative dose of nor-epinephrine in IE patients was almost 5 times that in the patients with VHD (p = 0.009).

## Measurement of cytokines and vasoactive peptides

### Acute phase regulation

Fig 4A shows that in the control group IL-6 was very low preoperatively and remained station-ary during the 1st 60 minutes of CPB. At the end of CPB its level increased to median (IQR) 24.48 pg/ml (40.09). In contrast to IE patients, the median IL-6 was already elevated preopera-tively (IE group: 25.0 (12.2–35.4), Control group: all below detection limit< = 9.2) and increased during CPB to duplicate after 60 minutes of CPB and reach five-fold its preoperative level at the end of CPB. IL-6 level at the end of CPB in the IE group was five-fold its level in the control group (119.73 pg/ml (226.49) vs. 24.48 pg/ml (40.09), $p = 0.001$)

In Fig 4B, the pre-operative level of CRP was significantly elevated preoperatively in IE patients (82.8 (24.1–118.7) vs. 3.1 (1.4–7.2), p<0.001) compared to control group. During CPB, the level of CRP fell slightly in both groups. Post-operatively, CRP increased steadily to reach maximal levels at 48 h.

In Fig 4C in both groups plasma level of PCT was low (<1 μg/l) before and during CPB and started to rise postoperatively. PCT was significantly higher in the IE group to the following time points: preoperatively (p = 0.017), during CPB (60min) (0.012) and after CPB (0.003).

### Biomarkers of cardiovascular dysfunction

Fig 5A shows that to all time points, MR-proADM was significantly higher in IE patients com-pared to control group. After starting CPB MR-proADM level increased in both groups to

A

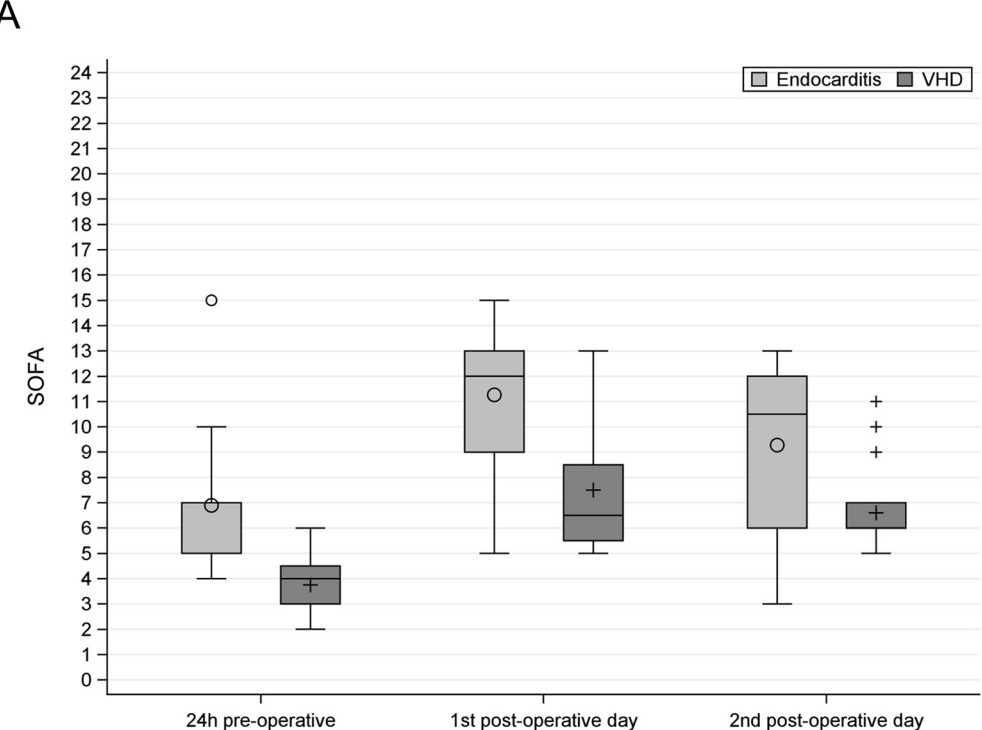

B

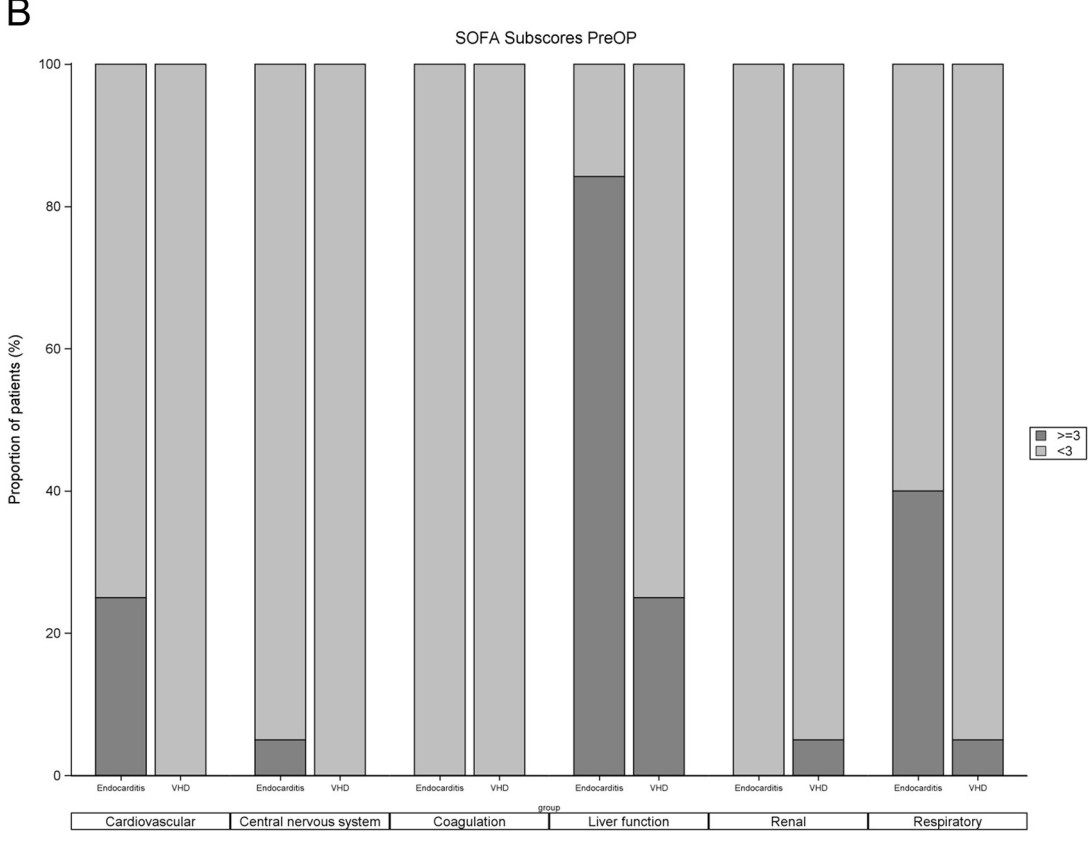

**Fig 2. A.** Boxplots comparing SOFA score of endocarditis patients (bright) to VHD patients (dark). SOFA: Sequential Organ Dysfunction; preop; within 24 h pre-operative; VHD: valvular heart disease. **B.** SOFA subscores within 24 h pre-operative in IE

and VHD patients. on the X-axis SOFA subscores within 24 h pre-operative in IE patients compared to control group. The dark columns represent the proportions of patients having organ failure (subscore ≥3), while the bright columns represent proportions of patients having no or less than sever organ dysfunctions (SOFA<3). SOFA: Sequential Organ Dysfunction; preop; within 24 h pre-operative; VHD: valvular heart disease.

reach highest levels 6 hours postoperative and started to decrease thereafter. Patients with IE had significantly higher levels of MR-proADM at each time point except at 24 and 48 h post-operative.

Fig 5B shows that pre-operative plasma level of CT-proAVP was slightly higher in IE group than the control group. After starting CPB, CT-proAVP levels in both groups increased to reach their maximum 6 h post- operatively (slightly higher in the VHD than the IE group). In the IE group its level decreased slower than in the VHD group (at 24 h and 48 h postoperative CT-proAVP was higher in IE group). To all time-point the differences in CT-proAVP levels between both groups were not statistically significant.

Fig 5C shows that plasma levels of CT-proET-1 to all time points were higher in IE patients. In both groups CT-proET-1 levels during CPB decreased to about 2/3 their preoperative levels and increased postoperatively to reach their maximum at 6 h post-operative. Plasma levels of CT-proET-1 in IE patients were significantly higher p 24 h preoperative, at the beginning of CPB and 60 minutes thereafter.

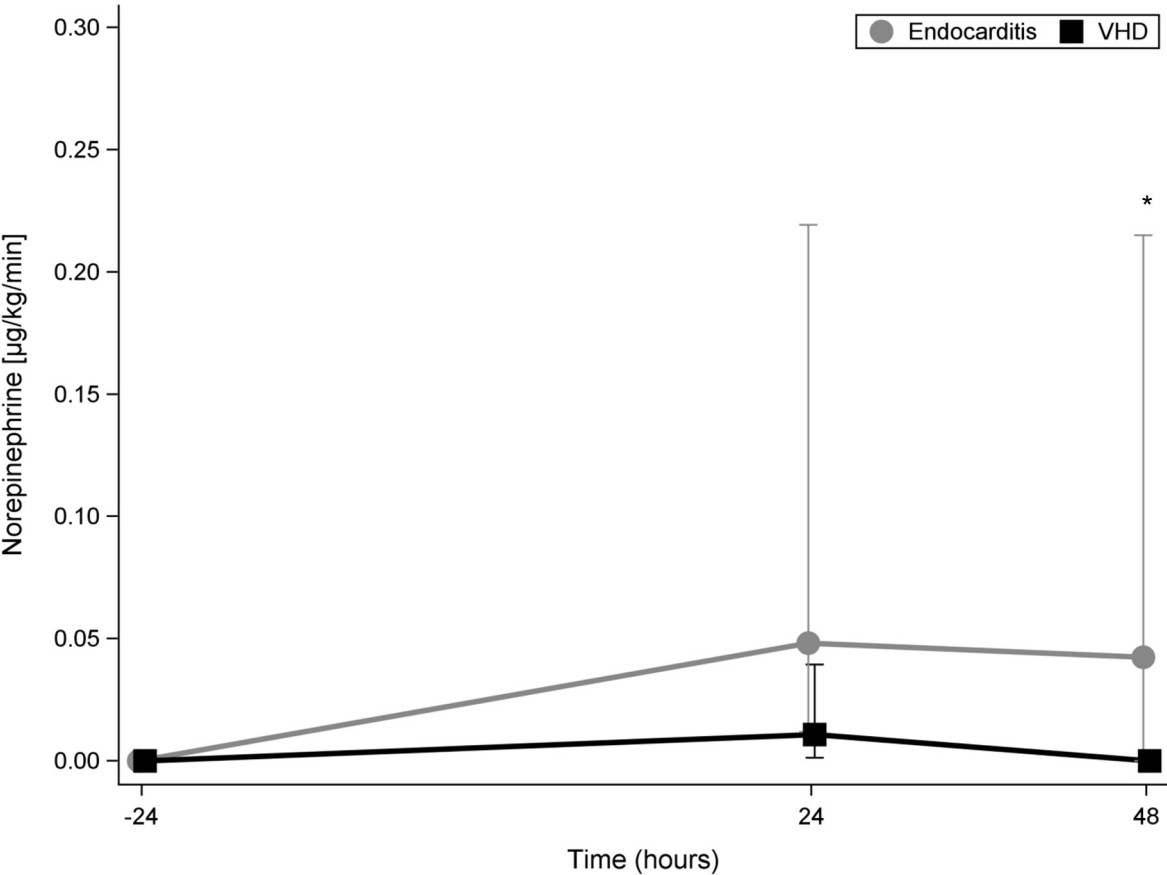

**Fig 3. Cumulative doses of norepinephrine in IE and in VHD patients.** The cumulative doses of norepinephrine administered during the 24 hours pre-operative (-24) as well as during the 1st post-operative day (24) and during the 2nd post-operative (48) day in patients with infective endocarditis (bright line) compared to VHD patients. VHD: valvular heart disease (dark line).

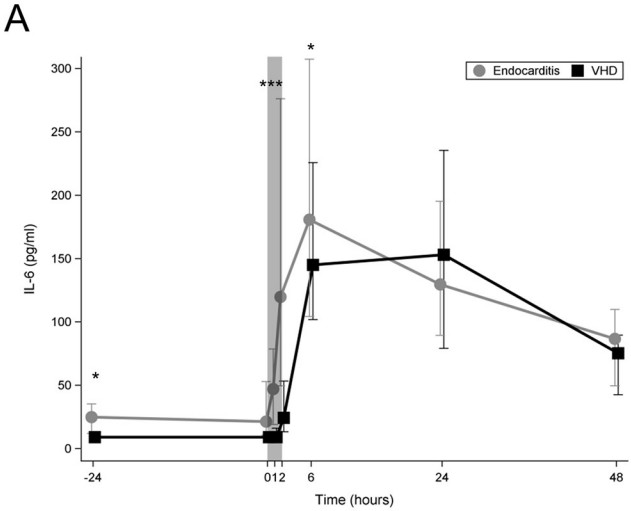

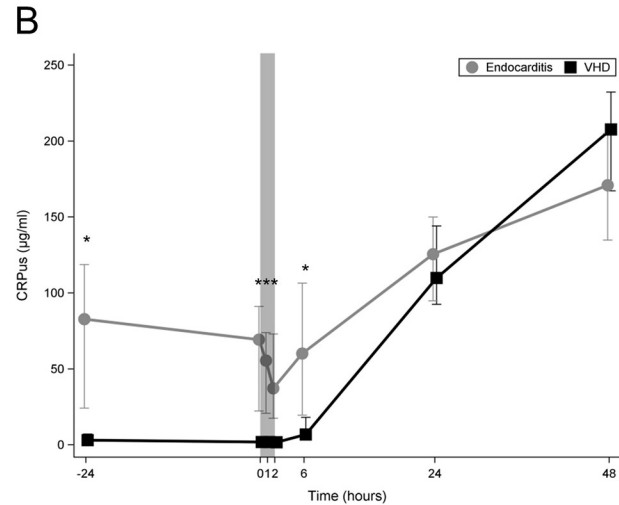

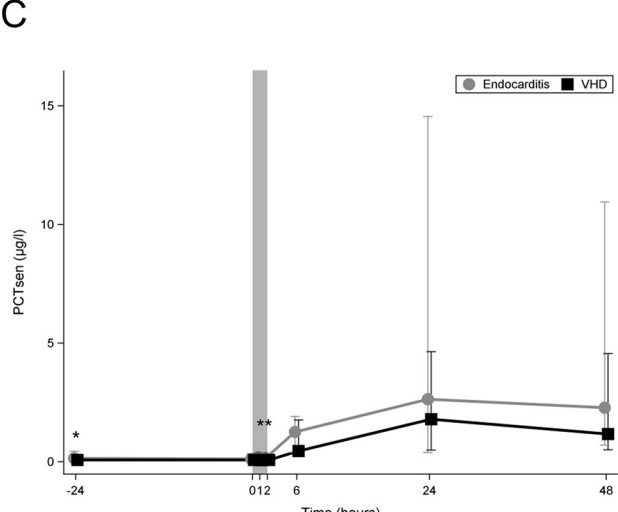

**Fig 4. Line chart comparing median plasma levels of markers of acute phase regulations between the two study groups.** (A) median plasma levels of IL-6 (B) median plasma levels of CRP, (C) median plasma levels of PCT.Bright line: patients with IE; dark line: patients with VHD; 0 on the x-axis represents the beginning of cardiopulmonary bypass (CPB); the shaded area represents the CPB time; $^*$: $p<0.05$; IE: infective endocarditis; VHD: valvular heart disease; IL: interleukin; CRP: C-reactive protein; PCT: procalcitonin.

Fig 5D shows that median MR-proANP before surgery was significantly higher in the IE group compared to the control group. After starting CPB the level of MR-proANP increased to reach its maximal level at the end of CPB in both groups and decreased thereafter. To all time-points except for at 24h and 48h post-operative, MR-proANP was significantly higher in the IE group.

## Inflammasome activation

Fig 6A shows that in both groups, pre-operative median plasma levels of IL-1β were below detection limit (< 2 pg/ml). In IE group after starting CPB median plasma level of IL-1β rapidly increased to reach a maximal level of 25.8 pg/ml and rapidly declined to approximately normal levels at the end of CPB. In the control group, IL-1β could not be detected at any time-point.

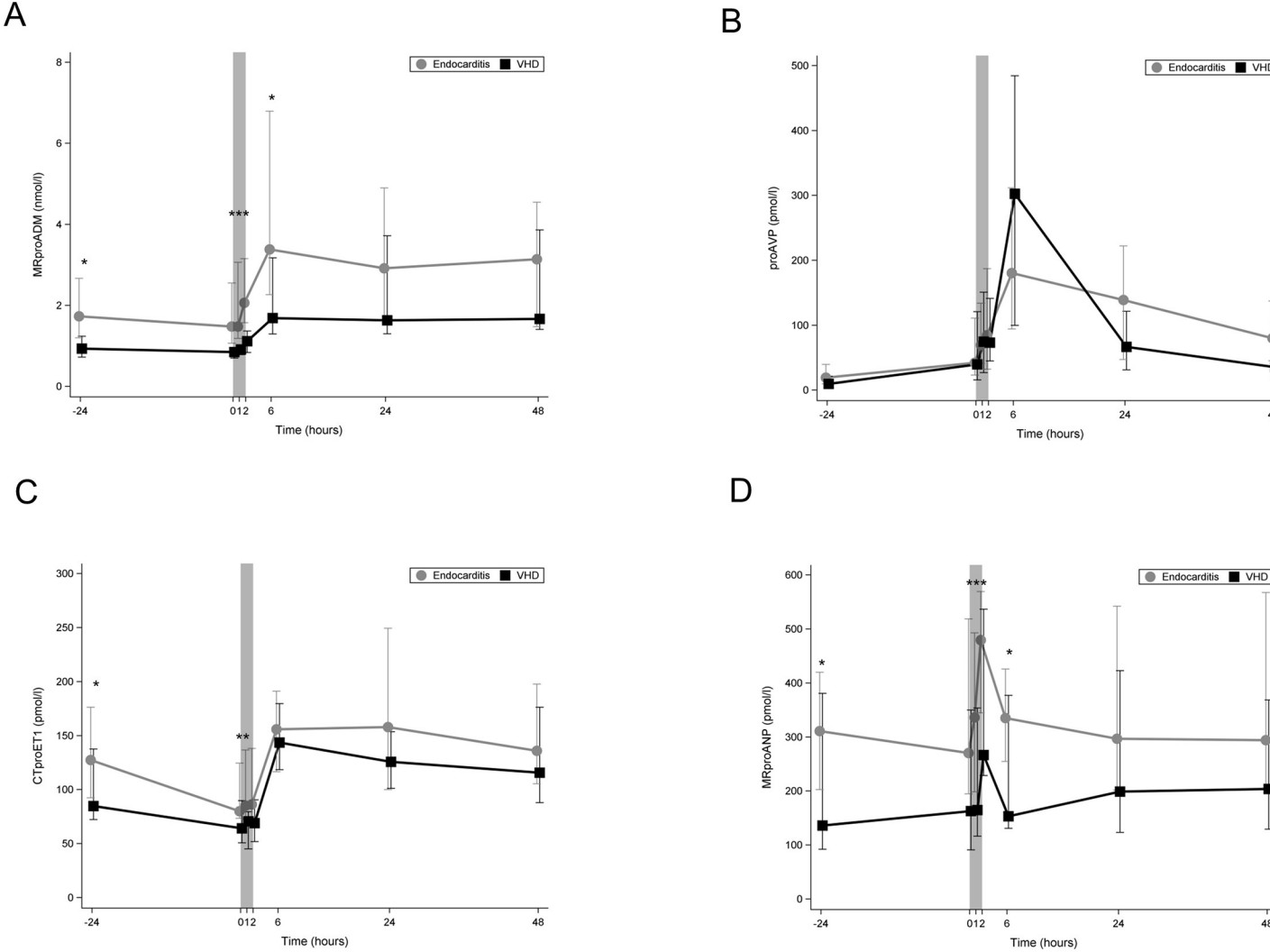

**Fig 5. Line chart comparing median plasma levels of biomarkers of cardiovascular dysfunction between the two study groups.** (A) Line chart comparing median plasma levels of MR-proADM. (B) Line chart comparing median plasma levels of proAVP. (C) Line chart comparing median plasma levels of CT-proET-1. (D) Line chart comparing median plasma levels of MR-proANP. Bright line: IE patients; dark line: VHD patients; 0 on the x-axis represents the beginning of cardiopulmonary bypass (CPB); the shaded area represents the CPB time; *: $p < 0.05$; IE: infective endocarditis; VHD: valvular heart disease; MR-proADM: midregional pro adrenomedullin; proAVP: copeptin pro vasopressin; CTproET-1: C-terminal pro endothelin; MR-proANP: midregional pro atrial natriuretic peptide.

Fig 6B shows that in both groups, Plasma levels of IL-18 steadily increased during CPB and continued to increase until 24 h postoperatively in the control group, while in the IE group it continued to increase until 48 h postoperatively. At all time-points, the plasma levels of IL-18 were significantly higher in IE group.

## Regulation of the inflammatory response

Fig 7A shows that in both groups TNF-α was not detectable preoperatively. In the IE group its level increased immediately after starting CPB while in the control group it remained undetected at each time point.

Fig 7B shows that plasma levels of IL-10 in both groups increased immediately after starting CPB to reach its maximum at its end. Postoperatively, the decrease in the level of IL-10 was faster in the control group and its levels fell to the preoperative level 24 hours postoperatively.

A

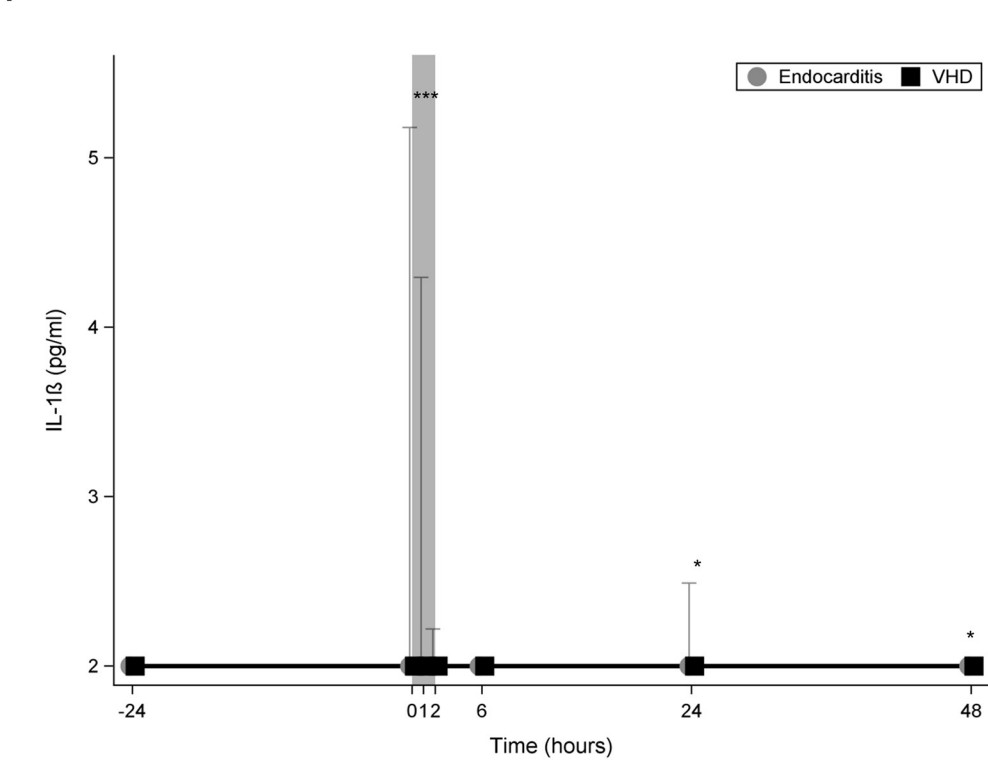

B

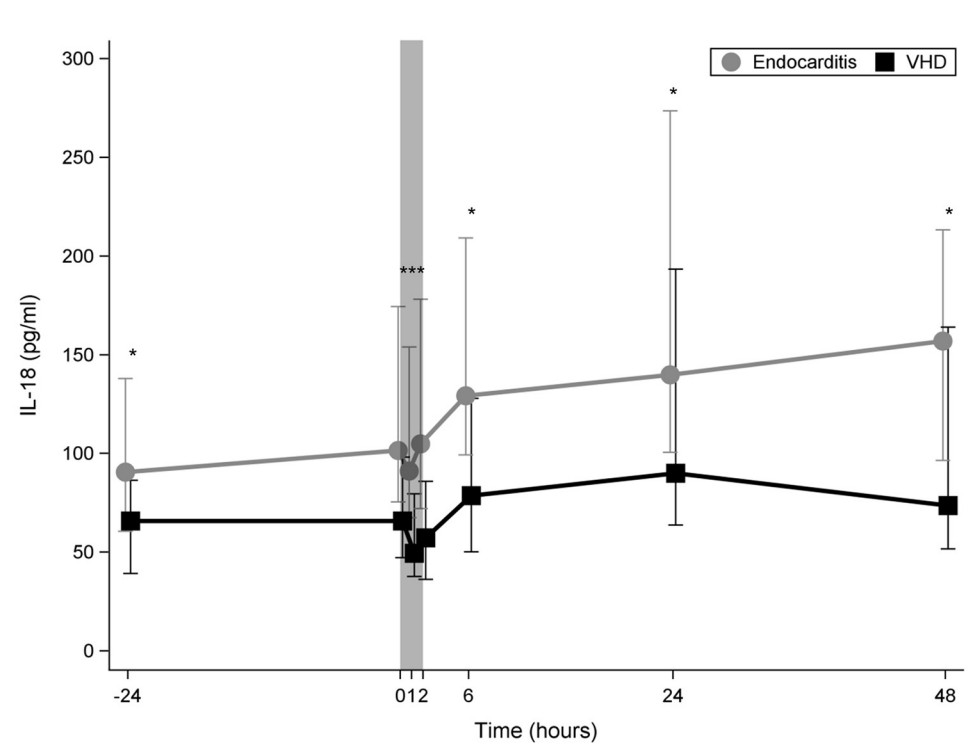

**Fig 6. Line chart comparing median plasma levels of markers of inflammasome activation between the two study groups.** (A) Line chart comparing median plasma levels ofIL-1β. (B) Line chart comparing median plasma levels of IL-

18. Bright line: IE; dark line: VHD; 0 on the x-axis represents the beginning of cardiopulmonary bypass (CPB); the shaded area represents the CPB time; *:$p < 0.05$; IE: infective endocadarditis; VHD: valvular heart disease; IL: interleukin.

In contrast, IL-10 in the IE group did not fall to the preoperative level even 48 h postoperatively.

Table in S1 Table shows the pre-operative details of the endocarditis patients.

Table in S2 Table shows the Spearman correlation analysis between the post-operative cumulative doses of vsopressor or catecholamine doses within 6 and 24h for the whole study population.

We found that MR-proANP correlated well with the post-operative cumulative doses of norepinephrine within 6h (rs 0.595, p<0.001) and within 24 hours (rs 0.602, p<0.001). MR-proADM correlated well with the post-operative cumulative doses of norepinephrine at 6h (rs 0.719, p<0.001) and 24h (rs 0.743, p<0.001). CT-proET1 correlated with the post-operative cumulative doses of epinephrine within 24h (rs 0.532, p = 0.0005).

Table 3 shows the microbiological profile of patients with IE. S. aureus was the most common pathogen (35%), followed by E. faecalis (15%). In 3 patients (15%), no pathogen could be identified.

## Marker level in survivors vs non-survivors

S1–S11 Figs Line charts of cytokines and vasoactive peptides at defined time points in non-survivors compared to survivors.

Non-survivors had significantly stronger acute phase reaction to CPB (higher IL-6 and CRP, and PCT levels (S1–S3 Figs) after starting CPB. Plasma levels of MR-proADM (S4 Fig) were also higher in non-survivors to almost all time points compared to survivors. The difference was statistically significant from the beginning of CPB until 6 hours post-operative. MR-proANP plasma levels (S7 Fig) were higher to all time points in non-survivors compared to survivors. The differences were statistically significant at 60 minutes after starting CPB until the 6 hours post-operative. IL-18 plasma levels (S9 Fig) to all time points were substantially higher in non-survivors compared to survivors and differences became statistically significant 60 minutes after starting CPB until 6 hours post-operative. Plasm levels of IL-10 increased similarly in non-survivors and survivors after starting CPB, however, it remained significantly higher to all post-operative time points in non-survivors compared to survivors (S11 Fig).

## Correlation of cytokines and vasoactive peptides (maximal level during CPB) and SOFA score on the 1st or 2nd postoperative day only for patients with IE

Table 4 shows the Spearman correlation analysis between the maximal level of cytokines and vasoactive peptides during CPB and SOFA score on the 1st and 2nd post-operative days for patients with IE. The maximal level of IL-6 during CPB correlated well with the SOFA score on the 1st and 2nd postoperative days (correlation coefficients (rs) 0.533, p = 0.019 and rs 0.574, p = 0.0127, respectively). The maximal level of PCT and MR-proADM correlated well with SOFA score on the 2nd postoperative day (rs 0.640, p = 0.005 and rs 0.630, p = 0.005, respectively).

We investigated the influence of surgical timing on mortality and on inflammatory markers. The mean time form diagnosis to operation was similar in survivors compared to non-survivors (6.62±12.78 vs 6.16±6.62 days, respectively). Table in S3 Table shows the influence of

A

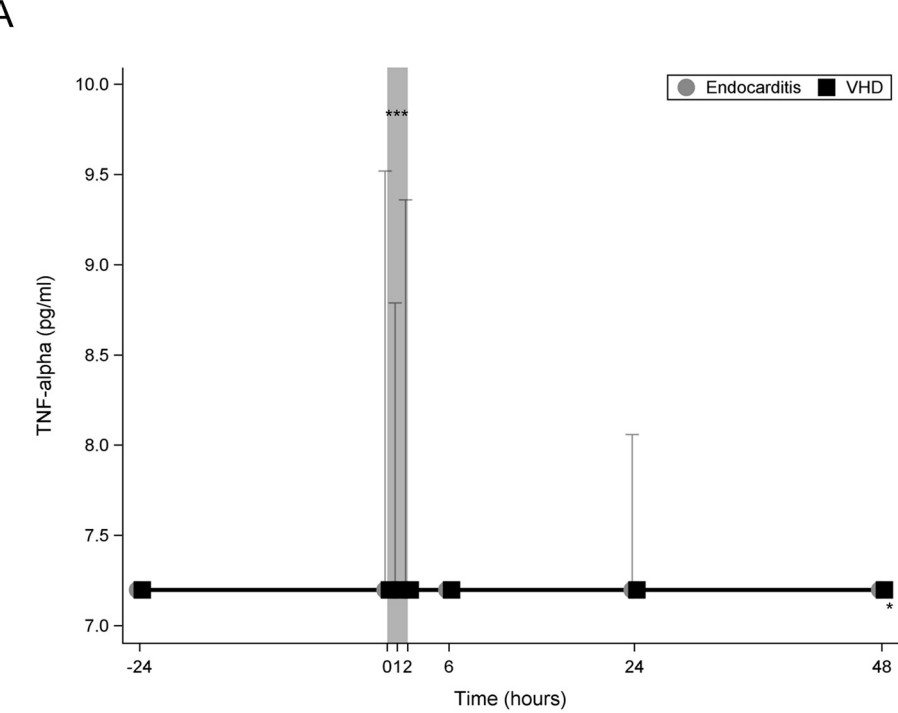

B

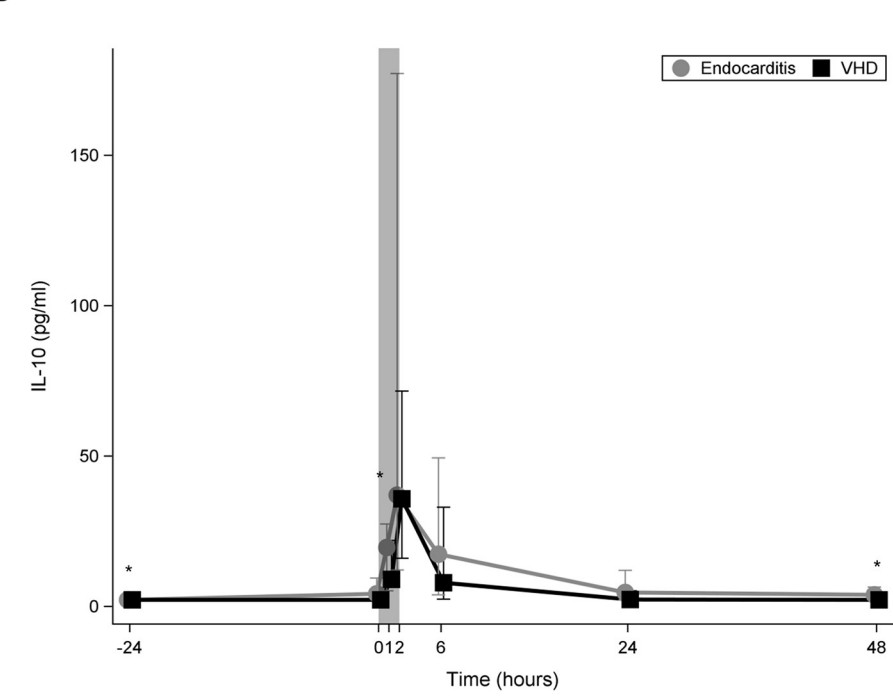

**Fig 7. Line chart comparing median plasma levels of regulators of the inflammatory response between the two study groups.** (A) Line chart comparing median plasma levels of TNF-alpha. (B) Line chart comparing median plasma levels of IL-10. Bright line: IE; dark line: VHD; 0 on the x-axis represents the beginning of cardiopulmonary bypass (CPB); the shaded area represents the CPB time; *:$p<0.05$; IE: infective endocarditis; VHD: valvular heart disease; TNF: tumour necrosis factor; IL: interleukin.

**Table 3. Shows the microbiological profile of patients with infective endocarditis.**

|  | Number (%) of patients |
|---|---|
| No findings | 3 (15.0%) |
| Staphylococcus aureus, MSSA | 7 (35.0%) |
| coagulase-negative Staphylococci CoNS | 3 (15.0%) |
| Enterococci (E. faecalis, E. faecium) | 3 (15.0%) |
| Streptococci | 3 (15.0%) |
| other gram-positive Cocci | 1 (5.0%) |
| Candida albicans | 1 (5.0%) |

MSSA: methicillin-susceptible Staphylococcus aureus; coagulase-negative staphylococci (CoNS).

surgical timing on cytokines and vasoactive peptide. There is a negative correlation between the time from diagnosis to surgery and IL-1 at 60 minutes CPB (rs -0.536, p = 0324), TNF-$\alpha$ at 24h post-operatively (rs -0.550, p = 0.0147), and IL-18 24h (rs -0.598, p = 0.007) and 48h post-operatively (rs -0.591, p = 0.010), which means the longer the time from the diagnosis to operation, the lower the level of aforementioned markers.

**Table 4. Spearman correlation analysis between the maximal level of cytokines and vasoactive peptides during cardiopulmonary bypass and SOFA score on the 1st and 2nd post-operative days for patients with IE.**

| Maximal level on CPB | SOFA 1st post-op. day (n = 19) | SOFA 2nd post-op. day (n = 18) |
|---|---|---|
| IL-6 rs | **0.533** | **0.574** |
| p | **0.019** | **0.0127** |
| C-reactive protein rs | -0.182 | 0.305 |
| p | 0.459 | 0.219 |
| Procalcitonin rs | 0.232 | **0.640** |
| p | 0.339 | **0.005** |
| MR-proANP rs | 0.235 | 0.212 |
| p | 0.333 | 0.398 |
| CT-proAVP rs | -0.026 | -0.040 |
| p | 0.916 | 0.875 |
| CT-proET1 rs | 0.232 | 0.439 |
| p | 0.339 | 0.068 |
| MR-proADM rs | 0.363 | **0.630** |
| p | 0.127 | **0.005** |
| IL-1$\beta$ rs | 0.300 | -0.141 |
| p | 0.199 | 0.576 |
| IL-18 rs | -0.039 | -0.160 |
| p | 0.8756 | 0.528 |
| TNF-$\alpha$ rs | -0.113 | -0.026 |
| p | 0.646 | 0.918 |
| IL-10 rs | 0.274 | 0.330 |
| p | 0.256 | 0.181 |

CPB: cardiopulmonary bypass; SOFA: Sequential Organ Failure Assessment score; rs: Spearman´s Rank Correlation Coefficient IL: inteleukin; MR-proANP: midregional pro adrenomedullin; MR-proANP: midregional pro atrial natriuretic peptide; CT-proAVP: copeptin midregional pro vasopressin; CT-proET1: C-terminal pro endothelin; TNF: tumor necrosis factor

We also investigated the correlation between the duration of antibiotic therapy and the level of different cytokines and vasoactive peptides. There is no correlation between the duration of antibiotic therapy and inflammatory markers as shown in S4 Table

## Discussion

Our results show that the presence of infective endocarditis during cardiac valve surgery is associated with increased inflammatory response as evident by higher plasma cytokine levels and other inflammatory mediators. Actively reducing inflammatory response appears to be a plausible therapeutic concept.

In our study, IE patients undergoing valvular surgery had significantly higher mortality and higher multiple organ dysfunction (MOD) than patients with non-infectious valvular heart disease (VHD). IE patients required more post-operative norepinephrine therapy compared to the control group. In addition, the intensity of the acute phase response (IL-6, CRP, and PCT) and inflammmasome activation (IL-1β) and the release of vasoactive peptides (MR-proADM and CT-proET-1) after starting the CPB were stronger in IE patients compared to the VHD patients. Plasma levels of certain cytokines; CRP, PCT, IL-6, IL-10, or IL-18 or vasoactive peptides; MR-proADM or MR-proANP were significantly higher in non-survivors compared to survivors. Among IE patients, during CPB there was a correlation between maximal levels of IL-6, MR-proANP, or PCT and the post-operative SOFA score. Thus, we conclude that the stronger inflammatory reaction observed in IE patients during CPB may be responsible for the higher MOD and higher in-hospital mortality observed in this group of patients undergoing cardiac surgery.

Surgical treatment is necessary in about 50% of IE patients and is associated with in-hospital mortality as high as 15–25% and 1-year mortality of 40% [1, 4]. The postoperative course of patients with IE is often complicated with a varying degree of circulatory failure i.e. hypotension, decreased systemic vascular resistance, despite high cardiac output, adequate fluid resuscitation, and adrenergic vasopressor administration which can progress to septic shock in up to 10–28% of cases [5–7]. The mechanism of development of circulatory failure after cardiac surgery in IE patients is still not well elucidated. CPB is known to induce systemic inflammation, which might lead to circulatory failure and MOD [14]. However, the incidences of MOD and mortality after cardiac surgery in IE patients are higher than in non-infectious VHD. A plausible explanation may be the difference in severity of inflammatory response to surgery between IE patients and VHD patients. Cytokines are regulators of the immune response to infection and play a key role in regulating inflammation and trauma. They can be used to measure the degree of inflammation [15]. To date, our study is the first study to investigate not only the pre-operative, but also the intra-, and post-operative levels of cytokines and vasoactive peptides in IE patients and in comparison to non-infectious VHD patients.

Bustamante et al., measured pre-operative levels of IL-6, IL-8, and IFN-γ in patients with prosthetic vale endocarditis and found that they were linked to mortality [16]. However, they did not measure intra- or post-operative cytokines levels. Several other studies have shown elevation of cytokines in the serum of IE patients compared to healthy individuals [11, 17, 18].

In our study, we found that preoperative levels of IL-6, CRP, PCT, IL-18, IL10, MRproADM, and CTproET1 were higher in the IE patients compared to the control group.

CPB initiates a systemic inflammatory response. Bernardi et al., reported that in their control group (patients undergoing elective cardiac surgery with CPB) the IL-6 was not detectable preoperatively and that median IL-6 (first quartile, third quartile) increased to 63.6 pg/ml (41.2, 154.9) at the end of CPB [19]. In our study, in the control group IL-6 was very low preoperatively and remained stationary during the first 60 minutes of CPB. At the end of CPB its level increased to median (IQR) 24.48 pg/ml (40.09). In contrast to IE patients, median IL-6

was already elevated preoperatively and increased during CPB to duplicate after 60 minutes of CPB and reach five-fold its preoperative level at the end of CPB. IL-6 level at the end of CPB in the IE group was five-fold its level in the control group, this confirms that the acute phase response to CPB was stronger in IE patients compared to patients with non-infectious VHD.

Kellum et al., have shown that in patients with sepsis, mortality was higher in patients with high level IL-6 [15]. In our study, the maximal level of IL-6 during CPB correlated well with the post-operative SOFA score. In addition, the median level of IL-6 in non-survivors was significantly higher than in survivors.

In our study, IL-18 could be detected pre-operatively in both groups, however, its level was higher in IE patients compared to the control group. Venkatachalam et al., have shown that IL-18 is a key pro-inflammatory mediator in the pathogenesis and deterioration of patients with heart and vascular disease [20]. This may explain the presence of IL-18 in the pre-operative measurements in both groups in our study. At the end of CPB, IL-18 level was higher than pre-operative level in the IE group, while in the control group its level was lower than pre-operative level. This may be explained by the inflammasome activation in the IE group. In our patients, IL-18 was significantly higher in non-survivors compared to survivors.

IL-1β is a cytokine that plays critical roles in inflammation and cardiac dysfunction during severe sepsis [21]. In our study, plasma level of IL-1β was not detectable before cardiac surgery in both groups. After starting CPB its level increased only in the IE group which suggests that inflammasome activation occurred only in IE patients.

## Vasoactive peptides

Elke et al., reported that high or increasing MR-proADM concentrations may help identify patients with a high risk of progression towards sepsis-related MOD[22]. Our results are in consistence with their results. In our study, plasma level of MR-proADM during CPB correlated well with Δ SOFA score. In addition, its level was significantly higher in non-survivors compared to survivors.

In both groups, plasma level of MR-proADM increased after starting CPB to reach its maximal level at 6 hours post-operative. Patients with IE had significantly higher levels of MR-proADM at each time point except at 24 and 48 h post-operative.

Increased CT-proAVP concentration is described in several studies as a strong predictor of mortality in patients with chronic as well as acute heart failure [23, 24]. CT-proAVP is the precursor peptide of proAVP and consequently AVP. The most important stimulus for AVP release is a change in plasma osmolality, a small change, of even 1%, in plasma osmolality is sufficient to change AVP concentration AVP is an important factor of the response and adaptation to stress[25]. In our study, preoperative level of CT-proAVP was slightly higher in IE group compared to control group. During CPB, CT-proAVP levels increased similarly in both groups. The stimulus for this increase may be the change of osmolality due to CPB. CT-proAVP reached its maximal level 6 hours post-operative in both groups.

Natriuretic peptides (NPs) include atrial natriuretic peptide (ANP), brain natriuretic peptide (BNP), and C-type NP[26]. ANP and BNP plasma levels are correlated with the degree of heart failure. MR-proANP has a significant diagnostic and prognostic utility in patients with heart failure [27]. MR-proANP has been shown as a predictor for mortality in patients with septic shock [28]. In our study, MR-proANP before surgery was significantly higher in the IE group compared to the control group. This was clinically reflected by the higher frequency of severe heart failure (NYHA ≥ III) in the IE group. After starting CPB the level of MR-proANP increased to reach its maximal level at the end of CPB in both groups and decreased thereafter. At almost each time point the level of MR-proANP in IE group was double that in the control

group. The maximal level of MR-proANP during HLM correlated well with the post-operative SOFA score in IE patients. In addition, plasma levels of MR-proANP were significantly higher in non-survivors compared to survivors.

Endothelin-1 is a very potent vasoconstrictor which has inotropic and pro-inflammatory properties. Plasma ET-1 are increased in patients with CHF [29] and in patients with septic shock [30] [31]. ET-1 itself, however, is difficult to measure due to its limited half-life. The precursor peptide C-terminal proendothelin-1 (CT-proET-1) is far more stable and allows a stoichiometric measurement of ET-1 [32]. Buendgens et al., showed that plasma levels of CT-proET-1 were higher in patients with sepsis compared to control group and that CT-proET-1 was an independent predictor of mortality [33]. In our study, CT-proET-1 was higher pre-operatively in IE patients. During CPB plasma levels of, CT-proET-1 decreased in both groups to about 2/3 their preoperative level. This could be explained with dilution caused by CPB. At 6 hours post-operative, its plasma levels reached their highest levels in both groups. The median CT-proET-1 in non-survivors was double that in survivors, however the difference was not statistically significant.

TNF-α is an essential component of the host immune response to infection and is responsible for the release of other pro- and anti-inflammatory mediators. TNF-α serum levels correlate with the severity of sepsis [34]. TNF-α plays a key role in the inflammatory response after CPB [35]. Excessive production of TNF-α may lead to organ dysfunction or death [36]. In our study, TNF-α was not detectable pre-operatively in both groups. After starting CPB, TNF-α plasma level increased only in the IE group and diminished rapidly after surgery.

IL-10 is one of the key anti-inflammatory cytokines as it decreases the production of inflammatory molecules, such as TNF-α and IL-6[37].

In our study, IL-10 in both groups increased after starting CPB to reach its maximum at its end. Postoperatively, the decrease in the level of IL-10 was faster in the control group and its levels fell to the pre-operative level 24 hours post-operatively. While in the IE group, IL-10 did not fall to the preo-perative level even 48 h post-operatively. Currently, there is a consensus that imbalance between anti- and pro-inflammatory cytokines e.g. IL-6 and IL-10 is one of the cause of mortality in sepsis [38]. Our results failed to confirm this theory, as in our patients IL-10 increased similarly in survivors and non-survivors after starting CPB to reach its peak at the end of it. Plasma level of IL-10 in survivors decreased rapidly to reach almost its pre-operative level at 6 h post-operative. In non-survivors, on the contrary, IL-10 remained significantly higher and diminished slower than in survivors. IL-6 was significantly higher in non-survivors at the end of CPB; however the difference declined gradually post-operative. Our results are consistent with those from a recently published study conducted on sepsis patients which showed that although IL-6 and IL-10 were associated with mortality, the balance between these two cytokines did not impact mortality[37].

In our study, patients with IE had higher operative risk, measured by EuroSCORE II, compared to patients with VHD. The expected mortalityin IE patients, based on EuroSCORE II, was 19% and the observed mortality was 35%. EuroSCORE II does not include liver dysfunction or failure in its calculation[39]. However, we and others showed that pre-operative liver failure is the strongest independent predictor of mortality in patients with IE [40, 41]. In addition, other factors such as the size of vegetation, abscess formation, S. *aureus* as causative pathogen, and the severity of preoperative neurological complications are independent predictors of mortality[42]. All these factors are not considered in EuroSCORE II.

## Limitations of the study

Cytokines secretion can be influenced by several factors (e.g., differences in techniques of surgery and anesthesia) which may lead to bias in comparing different patients groups. Although operative procedures were similar in both IE and in VHD group and despite standardized

operative and anesthetic techniques in our department, there might be some variations in techniques among surgeons and anesthetists. In addition, the limited number of patients is a partial limitation of this study.

## Conclusion

We could show in this pilot study that the inflammatory reaction to CPB, measured by different cytokines and vasoactive peptides, was stronger in patients with IE compared to patients with non-infectious VHD. We could also show that the magnitude of the inflammatory reaction correlated well with the degree of post-operative organ dysfunction. The results also showed that patients who died within 30 days of surgery had stronger inflammatory response than those who survived.

## Supporting information

**S1 Checklist. TREND statement checklist.**
(PDF)

**S1 Fig. Line chart comparing median plasma levels of IL-6 between survivors and non-survivors.** This is the S1 Fig legend: Bright line: non-survivors; dark line survivors; 0 on the x-axis represents the beginning of cardiopulmonary bypass (CPB); the shaded area represents the CPB time; IL: interleukin; *: $p < 0.05$.
(TIF)

**S2 Fig. Line chart comparing median plasma levels of C-reactive protein between survivors and non-survivors.** Bright line: non-survivors; dark line survivors; 0 on the x-axis represents the beginning of cardiopulmonary bypass (CPB); the shaded area represents the CPB time; CRP: C—reactive protein; *: $p < 0.05$.
(TIF)

**S3 Fig. Line chart comparing median plasma levels of procalcitonin between survivors and non-survivors.** Bright line: non-survivors; dark line survivors; 0 on the x-axis represents the beginning of cardiopulmonary bypass (CPB); the shaded area represents the CPB time; PCT: procalcitonin; *: $p < 0.05$.
(TIF)

**S4 Fig. Line chart comparing median plasma levels of MR-proADM between survivors and non-survivors.** Bright line: non-survivors; dark line survivors; 0 on the x-axis represents the beginning of cardiopulmonary bypass (CPB); the shaded area represents the CPB time; MR-proADM: midregional pro adrenomedullin; *: $p < 0.05$.
(TIF)

**S5 Fig. Line chart comparing median plasma levels of copeptin pro vasopressin between survivors and non-survivors.** Bright line: non-survivors; dark line survivors; 0 on the x-axis represents the beginning of cardiopulmonary bypass (CPB); the shaded area represents the CPB time; proAVP: copeptin pro vasopressin; *: $p < 0.05$.
(TIF)

**S6 Fig. Line chart comparing median plasma levels of CT-proET-1 between survivors and non-survivors.** Bright line: non-survivors; dark line survivors; 0 on the x-axis represents the beginning of cardiopulmonary bypass (CPB); the shaded area represents the CPB time; CT-proET-1: C-terminal pro endothelin-1.
(TIF)

**S7 Fig. Line chart comparing median plasma levels of MR-proANP between survivors and non-survivors.** Bright line: non-survivors; dark line survivors;0 on the x-axis represents the beginning of cardiopulmonary bypass (CPB); the shaded area represents the CPB time; MR-proANP: midregional pro atrial natriuretic peptide; *: $p < 0.05$.
(TIF)

**S8 Fig. Line chart comparing median plasma levels of IL-1β between survivors and non-survivors.** Bright line: non-survivors; dark line survivors; 0 on the x-axis represents the beginning of cardiopulmonary bypass (CPB); the shaded area represents the CPB time; IL: interleukin.
(TIF)

**S9 Fig. Line chart comparing median plasma levels of IL-18 between survivors and non-survivors.** Bright line: non-survivors; dark line survivors; 0 on the x-axis represents the beginning of cardiopulmonary bypass (CPB); the shaded area represents the CPB time; IL: interleukin;*: $p < 0.05$.
(TIF)

**S10 Fig. Line chart comparing median plasma levels of TNF-alpha between survivors and non-survivors.** Bright line: non-survivors; dark line survivors;0 on the x-axis represents the beginning of cardiopulmonary bypass (CPB); the shaded area represents the CPB time; TNF: tumour necrosis factor;*: $p < 0.05$.
(TIF)

**S11 Fig. Line chart comparing median plasma levels of IL-10 between survivors and non-survivors.** Bright line: non-survivors; dark line survivors; 0 on the x-axis represents the beginning of cardiopulmonary bypass (CPB); the shaded area represents the CPB time; IL: interleukin;*: $p < 0.05$.
(TIF)

**S1 File. Study protocol.**
(DOCX)

**S1 Table. Pre-operative characteristics of patients with infective endocarditis.**
(RTF)

**S2 Table. Spearman correlation analysis between the maximal level of inflammatory markers during CPB and cumulative doses of vasopressor or catecholamine doses at 6 and 24h postoperatively.**
(DOCX)

**S3 Table. Spearman correlation analysis between the time from diagnosis of infective endocarditis to operation and the levels of cytokines and vasoactive peptides at defined time points.**
(DOCX)

**S4 Table. Spearman correlation analysis between the duration of antibiotic therapy and the levels of cytokines and vasoactive peptides.**
(DOCX)

**S5 Table. Comparison of cytokines and vasoactive peptides between patients operated in minimally invasive technique compared to those operated in sternotomy within the VHD group.**
(DOCX)

## Acknowledgments

We wish to thank Mr. Ingo Curdt and Mr. Manne Krop for their technical support on the measurement of Cytokines and vasoactive peptides. We also wish to thank Mr. Vladimir Patchev and Mrs. Cornelia Eichhorn for project and data management.

## Author Contributions

**Conceptualization:** Mahmoud Diab, Christoph Sponholz, Mathias W. Pletz, Michael Bauer, Frank M. Brunkhorst, Torsten Doenst.

**Data curation:** Mahmoud Diab, Raphael Tasar, Christoph Sponholz.

**Formal analysis:** Mahmoud Diab, Thomas Lehmann.

**Funding acquisition:** Frank M. Brunkhorst.

**Investigation:** Mahmoud Diab, Raphael Tasar, Mathias W. Pletz, Frank M. Brunkhorst.

**Methodology:** Mahmoud Diab, Thomas Lehmann, Mathias W. Pletz, Michael Bauer, Torsten Doenst.

**Project administration:** Mahmoud Diab, Thomas Lehmann.

**Supervision:** Mahmoud Diab, Torsten Doenst.

**Validation:** Mahmoud Diab.

**Visualization:** Mahmoud Diab.

**Writing – original draft:** Mahmoud Diab, Raphael Tasar, Christoph Sponholz.

**Writing – review & editing:** Mahmoud Diab, Christoph Sponholz, Thomas Lehmann, Mathias W. Pletz, Michael Bauer, Frank M. Brunkhorst, Torsten Doenst.

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
