## [Decision Letter · Decision Letter 0]

8 Oct 2019

PONE-D-19-17603

Changes in Inflammatory and Vasoactive Mediator Profiles During Valvular Surgery With or Without Infective Endocarditis: A Case Control Pilot Study

PLOS ONE

Dear Dr. Diab,

Thank you for submitting your manuscript to PLOS ONE. After careful consideration, we feel that it has merit but does not fully meet PLOS ONE’s publication criteria as it currently stands. Therefore, we invite you to submit a revised version of the manuscript that addresses the points raised during the review process.

We would appreciate receiving your revised manuscript by Nov 22 2019 11:59PM. To enhance the reproducibility of your results, we recommend that if applicable you deposit your laboratory protocols in protocols.io, where a protocol can be assigned its own identifier (DOI) such that it can be cited independently in the future. For instructions see: http://journals.plos.org/plosone/s/submission-guidelines#loc-laboratory-protocols

We look forward to receiving your revised manuscript.

Kind regards,

Andrea Ballotta

Academic Editor

PLOS ONE

**Journal Requirements:**

2. Please provide additional details regarding participant consent. In the ethics statement in the Methods and online submission information, please ensure that you have specified what type you obtained (for instance, written or verbal, and if verbal, how it was documented and witnessed).

**Additional Editor Comments (if provided):**

Thank you for having submitted this paper. The manuscript is of sure interest but it needs major revisions as stated by the reviewers. So invite you

to revise promptly the paper and resubmit again.

**Comments to the Author**

1. Is the manuscript technically sound, and do the data support the conclusions?

Reviewer #1: Yes

Reviewer #2: Partly

2. Has the statistical analysis been performed appropriately and rigorously? 

Reviewer #1: Yes

Reviewer #2: Yes

3. Have the authors made all data underlying the findings in their manuscript fully available?

Reviewer #1: Yes

Reviewer #2: No

4. Is the manuscript presented in an intelligible fashion and written in standard English?

Reviewer #1: Yes

Reviewer #2: No

5. Review Comments to the Author

Reviewer #1: This is a nice and interesting study about a tricky pathophysiological condition.

Studying and profiling the "systemic scenario" of infective endocarditis and its potential operative and clinical consequenses is very challenging.

In this specific investiagion, authors state that in presence of infective endocarditis during cardiac valve surgery, inflammatory response is much stronger than the one experienced by patients with non-infectious valvular heart disease.

Even if cofounding factors can be numerous with such a background (some of them properly listed in limitations), this study could represent a meaningful trigger for further investigations, which are needed.

Methodological process is well written and results are clearly presented.

I would just suggest to provide some considerations about the potential role of surgical timing, combined with the use of antibiotics in limiting (or worsening) the inflammatory response.

Secondly, I would better underline the impact of the use of vasopressors in perpetuating cytokines release involved in microcirculatory dysfunction.

Finally, the limited amount of patients shoudl be mentioned as partial limitation as well.

Reviewer #2: The article presents a prospective case-control analysis of 20 patients with endocarditis vs 20 pts with VHD without endocarditis having undergone CPB for cardiac valve surgery from May to December 2016 either isolated or combined.

Endpoints:

1. Plasma profiles of inflammatory biomarkers at pre-defined time points during the surgical intervention were measured: Procalcitonin, C-reactive protein, C-terminal proendothelin-1(CT-peoET-1), tumor necrosis factor alpha (TNFα), interleukin (IL)-1β, IL-6, IL-10, IL-18.

2. Plasma profiles of inflammation-related vasoactive mediators at pre- defined time points the course of the surgical intervention were: midregional pro-Adrenomedullin (MR6 proADM), copeptin pro-Arginine Vasopressin (CT-117 proAVP), midregional pro-Atrial natriuretic Peptide (MR-proANP).

3. Changes in organ dysfunction during the 1st and 2nd post-operative days, disclosed by ΔSOFA score as compared to pre-surgery status.

4. Use and duration of renal replacement therapy.

5. Cumulative doses of concomitant medications (vasopressors, corticoids, prostaglandins) applied during the surgery and over 48 h thereafter.

6. In-hospital mortality within 30 days post-surgery.

For the study the authors collected preoperative, operative and postoperative data and blood samples for biomarkers at predefined times: - 12 to 24 hours before transfer to the operating theatre; - At connection to the CPB, - 60 minutes after connection to the CPB, - disconnection of the CPB - 6, 24 and 48 hours after the end of the operation.

The two groups were significantly different for EuroSCORE II, SOFA score, COPD, PAVD, Admission type (all VHD were ordinary, endocarditis were urgent (75%) or emergency (25%).

The results showed a lower time of CPB in VHD group (no X clamp time is reported). Mortality was significantly higher in endocarditis group (35 vs 5%). Postoperative hemodialysis was similar.

After that the authors reported all the figure legends and their results about the infiammatory results in a quite mechanical and confused way, please reorganize it and try to give us a more clear section.

Also in the discussion section a lot of information about each cytokines and inflammatory markers in my opinion can be cut to make the speech lighter and more fluent.

In the conclusions the authors state the inflammatory reaction to CPB was stronger in patients with IE group and that the magnitude of the inflammatory reaction correlated well with the degree of post-operative organ dysfunction.

Some comments.

Comment 1: the first two endpoints seem to be more a method more than an outcomes.

Comment 2: is not very clear how the patients with endocarditis have a so high Euroscore. The authors has to better explain the patients preoperative status, surgical indication (emergency operation at least) and their preparation to operation in term of ABT weeks, blood cultures, time from diagnosis and embolic stroke (?) (30% neurological disorders (!)). Moreover ther is a very high SD for the ES II of endocarditis group, in my opinion at least a comment of this findings is mandatory.

Comment 3: In the operative data is not very clear the type of surgery for each patients. I suggest to divide the pts for the primary indication (mitral or aortic surgery and to add the concomitant surgery). No one patients needed a Bentall operation? Which prosthesis were used? All pts underwent full sternotomy? Please add Xclamp time.

Comment 4: the mortality is quite high, probably a more specific explanation on the final cause of death can help us to understand the postoperative course; in my opinion is not very clear.

Comment 5: In my opinion the conclusions are not very clear because is not well explained the correlation between clinical deterioration and inflammatory markers. I mean is not very clear which is the authors indication after their findings and what can we clinical improve in terms of indication and timing. Or at least just add a comment.

6. PLOS authors have the option to publish the peer review history of their article (what does this mean?). If published, this will include your full peer review and any attached files.

Reviewer #1: No

Reviewer #2: No

---

## [Author Response · Author response to Decision Letter 0]

9 Dec 2019

Manuscript PONE-D-19-17603 Diab et al.

Response to the Reviewers' Comments

We would like to thank all reviewers for the constructive and helpful comments. We have taken care to revise the manuscript according to the critiques. We addressed below each comment point by point:

We deposited our protocol in protocols.io with the following DOI: 

dx.doi.org/10.17504/protocols.io.9esh3ee

Journal Requirements:

When submitting your revision, we need you to address these additional requirements

Response: we checked that our manuscript meets PLOS ONE´s style requirement

Please provide additional details regarding participant consent. In the ethics statement in the Methods and online submission information, please ensure that you have specified what type you obtained (for instance, written or verbal, and if verbal, how it was documented and witnessed).

Response:

We clarified that in page 5 in the manuscript with track changes

Obtaining the informed consent:

A written informed consent was obtained from each patient before inclusion in the study according to § 28 of the Declaration of Helsinki.

Reviewer #1

We wish to thank the reviewer for the kind and favorable comments

This is a nice and interesting study about a tricky pathophysiological condition.

Studying and profiling the "systemic scenario" of infective endocarditis and its potential operative and clinical consequences is very challenging.

In this specific investigation, authors state that in presence of infective endocarditis during cardiac valve surgery, inflammatory response is much stronger than the one experienced by patients with non-infectious valvular heart disease.

Even if cofounding factors can be numerous with such a background (some of them properly listed in limitations), this study could represent a meaningful trigger for further investigations, which are needed.

Methodological process is well written and results are clearly presented.

Comment 1: I would just suggest to provide some considerations about the potential role of surgical timing, combined with the use of antibiotics in limiting (or worsening) the inflammatory response.

Response: We investigated the influence of surgical timing on survival and on inflammatory response. 

The mean time form diagnosis to operation was similar in survivors compared to non-survivors (6.62±12.78 vs 6.16±6.62 days, respectively). There is a negative correlation between the time from diagnosis to surgery and IL-1 at 60 minutes CPB (Spearman´s Rank Correlation Coefficient (rs) -0.536, p=0324), TNF-α at 24h post-operatively (rs -0.550, p= 0.0147), and IL-18 24h (rs -0.598, p= 0.007) and 48h postoperatively (rs -0.591, p=0.010), which means the longer the time from the diagnosis to operation, the lower the level of aforementioned markers.

We also investigated the correlation between the duration of antibiotic therapy and the level of different cytokines and vasoactive peptides. There is no correlation between the duration of antibiotic therapy and inflammatory markers. 

We added this information to the results section of the manuscript page 20 of the track changes manuscript and provided 2 tables S3 Table and S4 Table with the details of the Spearman correlation analyses for timing of surgery and for duration of antibiotic therapy as supporting information.

Secondly, I would better underline the impact of the use of vasopressors in perpetuating cytokines release involved in microcirculatory dysfunction.

Response: To clarify this point, we performed Spearman correlation analyses between the cumulative doses of vasopressor or catecholamine doses at 6 and 24h postoperatively. 

We found that MR-proANP correlated well with the post-operative cumulative doses of norepinephrine within 6h (rs 0.595, p<0.001) and within 24 hours (rs 0.602, p<0.001). MR-proADM correlated well with the postoperative cumulative doses of norepinephrine at 6h (rs 0.719, p<0.001) and 24h (rs 0.743, p<0.001). CT-proET1 correlated with the post-operative cumulative doses of epinephrine within 24h (rs 0.532, p=0.0005). We added this text to page 17 of the manuscript with track changes and submitted S2 Table with details of this analysis. 

Finally, the limited amount of patients should be mentioned as partial limitation as well.

Response: We added the limited number of patients as partial limitation in page 26 of the manuscript with track changes.

Reviewer #2: 

We wish to thank the reviewer for the constructive comments

The article presents a prospective case-control analysis of 20 patients with endocarditis vs 20 pts with VHD without endocarditis having undergone CPB for cardiac valve surgery from May to December 2016 either isolated or combined.

Endpoints:

1. Plasma profiles of inflammatory biomarkers at pre-defined time points during the surgical intervention were measured: Procalcitonin, C-reactive protein, C-terminal proendothelin-1(CT-peoET-1), tumor necrosis factor alpha (TNFα), interleukin (IL)-1β, IL-6, IL-10, IL-18.

2. Plasma profiles of inflammation-related vasoactive mediators at pre- defined time points the course of the surgical intervention were: midregional pro-Adrenomedullin (MR6 proADM), copeptin pro-Arginine Vasopressin (CT-117 proAVP), midregional pro-Atrial natriuretic Peptide (MR-proANP).

3. Changes in organ dysfunction during the 1st and 2nd post-operative days, disclosed by ΔSOFA score as compared to pre-surgery status.

4. Use and duration of renal replacement therapy.

5. Cumulative doses of concomitant medications (vasopressors, corticoids, prostaglandins) applied during the surgery and over 48 h thereafter.

6. In-hospital mortality within 30 days post-surgery.

For the study the authors collected preoperative, operative and postoperative data and blood samples for biomarkers at predefined times: - 12 to 24 hours before transfer to the operating theatre; - At connection to the CPB, - 60 minutes after connection to the CPB, - disconnection of the CPB - 6, 24 and 48 hours after the end of the operation.

The two groups were significantly different for EuroSCORE II, SOFA score, COPD, PAVD, Admission type (all VHD were ordinary, endocarditis were urgent (75%) or emergency (25%).

The results showed a lower time of CPB in VHD group (no X clamp time is reported). Mortality was significantly higher in endocarditis group (35 vs 5%). Postoperative hemodialysis was similar.

After that the authors reported all the figure legends and their results about the infiammatory results in a quite mechanical and confused way, please reorganize it and try to give us a more clear section.

Response: We changed the figure titles and legends to meet the requirements of the PLOS ONE guidelines. Captions are not allowed to be included as part of the figure file or in a separate document.

Also in the discussion section a lot of information about each cytokines and inflammatory markers in my opinion can be cut to make the speech lighter and more fluent.

Response: We cut the unnecessary information in the discussion which does not serve our argument. 

In the conclusions the authors state the inflammatory reaction to CPB was stronger in patients with IE group and that the magnitude of the inflammatory reaction correlated well with the degree of post-operative organ dysfunction.

Some comments.

Comment 1: the first two endpoints seem to be more a method more than an outcomes.

Response: We changed the text to:

1. Plasma profiles of inflammatory biomarkers: Procalcitonin, C-reactive protein, C-terminal proendothelin-1(CT-peoET-1), tumor necrosis factor alpha (TNFα), interleukin (IL)-1β, IL-6, IL-10, IL-18. 

2. Plasma profiles of inflammation-related vasoactive mediators: midregional pro-Adrenomedullin (MR-proADM), copeptin pro-Arginine Vasopressin (CT-proAVP), midregional pro-Atrial natriuretic Peptide (MR-proANP)

please see page 5 Endpoints in the main manuscript with track changes.

Comment 2: is not very clear how the patients with endocarditis have a so high Euroscore. The authors has to better explain the patients preoperative status, surgical indication (emergency operation at least) and their preparation to operation in term of ABT weeks, blood cultures, time from diagnosis and embolic stroke (?) (30% neurological disorders (!)). Moreover ther is a very high SD for the ES II of endocarditis group, in my opinion at least a comment of this findings is mandatory.

Response: EuroSCORE II was high in the IE group because:

• 30% of IE patients had previous cardiac surgery compared to 5% in VHD group

• 75% of the IE patients underwent urgent operation and the rest underwent emergency operation, while all patients in VHD group were operated electively.

• 50% of patients with IE were poorly mobile preoperatively while only 5% of patients in VHD group were poorly mobile.

• 60% of IE patients had a critical pre-operative state compared to only one patient in the VHD group. 

• In addition, the presence of IE per se increases the EuroSCORE II. 

We added the factors we mentioned above to the pre-operative patient characteristics presented in Table 1 page 10 in the main manuscript with track changes.

In addition we added a table with more pre-operative details of the endocarditis patients:S1 Table. 

We investigated the influence of surgical timing on survival and on inflammatory response. 

The mean time form diagnosis to operation was similar in survivors compared to non-survivors (6.62±12.78 vs 6.16±6.62 days, respectively). There is a negative correlation between time from diagnosis to surgery and IL-1 at 60 minutes CPB (Spearman´s Rank Correlation Coefficient (rs) -0.536, p=0324), TNF-α at 24h post-operative (rs -0.550, p= 0.0147), and IL-18 24h (rs -0.598, p= 0.007) and 48h postoperative (rs -0.591, p=0.010), which means the longer the time from the diagnosis to operation, the lower the level of aforementioned markers.

We also investigated the correlation between the duration of antibiotic therapy and the level of different cytokines and vasoactive peptides. There is no correlation between the duration of antibiotic therapy and inflammatory markers. 

We added this information to the results section of the manuscript and provided 2 tables S3 Table and S4 Table with the details of the Spearman correlation analyses for timing of surgery and for duration of antibiotic therapy as supporting information.

We agree that the SD of the EuroSCORE II is much higher in the endocarditis group than in the VHD group. However, in order to compare the dispersion of the groups the coefficient of variance (CV) should be used, which is a standardized measure (SD divided by the mean). Since the mean is larger in the Endocarditis group than in the VHD group, the CVs are not much different (CV Endocarditis=0.94, CV VHS=0.80). In addition, the greater standard deviation also reflects the expected greater heterogeneity of endocarditis patients compared to an elective patient population undergoing valve surgery. 

Comment 3: In the operative data is not very clear the type of surgery for each patient. I suggest to divide the pts for the primary indication (mitral or aortic surgery and to add the concomitant surgery). No one patient needed a Bentall operation? Which prosthesis were used? All pts underwent full sternotomy? Please add Xclamp time.

Response: We added to Table 2 which shows Operative data and outcome in page 11 of the main manuscript with track changes information about the type of surgery in each group of patients, type of prostheses used, X-clamp time and the approach. 

We found that 55% of patients in the VHD group underwent minimally invasive surgery compared to only 15% in the endocarditis. This difference might cause bias in comparing the inflammatory reaction between both groups. In order to investigate whether the minimally invasive approach influences the inflammatory reaction or not, we compared the cytokine release in patients who underwent minimally invasive surgery compared to those who underwent full sternotomy within the VHD group. We found that for all markers to all time points, there was no significant difference between patient operated via minimally invasive approach and those operated via sternotomy, except for the median IL-10 at 24h postoperative which was higher in the sternotomy group (3.7 (IQR 6.32) vs. 2.20 (IQR 0.27). We added S5 Table to the supporting information showing the median and IQR of all markers to the defined time points in patients who underwent full sternotomy compared to patients operated via minimally invasive approach within the VHD group.

In our study population no patient underwent a Bentall operation.

Comment 4: the mortality is quite high, probably a more specific explanation on the final cause of death can help us to understand the postoperative course; in my opinion is not very clear.

Response: In our study the cause of mortality in all non-survivors was post-operative multiple organ failure.

As we have shown in Table 1, patients with IE had high EuroSCORE II 18.6�17.4 and high pre-operative SOFA score 6.895�2.58 which reached 11.263�2.579 on the 1st postoperative day. It has been previously shown that critically ill patients with SOFA score between 7-9 had a mortality of around 20%, and in patients who reached SOFA score 10-12 the mortality rateincreased to 40- 50%(1, 2). Thus, in our high risk group of IE patients it was expected to have such a high mortality. Despite the expected high mortality, surgery is still recommended for such patients. We have previously shown that the in-hospital mortality in endocarditis patients who did not underwent cardiac surgery despite having an indication was 65%(3).

.

 Comment 5: In my opinion the conclusions are not very clear because is not well explained the correlation between clinical deterioration and inflammatory markers. I mean is not very clear which is the authors indication after their findings and what can we clinical improve in terms of indication and timing. Or at least just add a comment.

Response: We could show in this pilot study that the inflammatory reaction to CPB, measured by different cytokines and vasoactive peptides, was stronger in patients with IE compared to patients with non-infectious VHD. This conclusion was based on the following findings:

• We have illustrated this in Fig 4A that IL-6 level at the end of CPB in the IE group was five-fold its level in the VHD group (119.73 pg/ml (226.49) vs. 24.48 pg/ml (40.09), p= 0.001). 

• I Fig 5A: Patients with IE had significantly higher levels of MR-proADM at each time point except at 24 and 48 h post-operative.

• In Fig 5C Plasma levels of CT-proET-1 in IE patients were significantly higher p 24 h preoperative, at the beginning of CPB and 60 minutes thereafter

• In Fig 5D: MR-proANP was significantly higher in the IE group during CPB and at 6h post-operatively

• In Fig 6A In IE group after starting CPB median plasma level of IL-1β rapidly increased to reach a maximal level of 25.8 pg/ml and rapidly declined to approximately normal levels at the end of CPB. In the control group, IL-1β could not be detected at any time-point.

• In Fig 7A: we showed that in both groups TNF-α was not detectable preoperatively. In the IE group its level increased immediately after starting CPB while in the control group it remained undetected at each time point.

We could also demonstrate in Table 4 that the increase of levels of IL-6, PCT, and MR-proADM during CPB was associated with high SOFA score post-operatively. Thus, we can conclude that the magnitude of the inflammatory reaction correlated well with the degree of post-operative organ dysfunction. As SOFA score is the best score to report organ dysfunctions in sepsis patients as recommended by the Task Force for sepsis (4), we consider SOFA score an accurate tool to measure and report clinical deterioration. 

In addition, we also demonstrated in supporting information S1-11 Figs that Non-survivors had significantly stronger acute phase reaction to CPB (higher IL-6 and CRP, and PCT levels (S1-3 Fig) after starting CPB. Plasma levels of MR-proADM (S4 Fig) were also higher in non-survivors to almost all time points compared to survivors. The difference was statistically significant from the beginning of CPB until 6 hours post-operative. MR-proANP plasma levels (S7 Fig) were higher to all time points in non-survivors compared to survivors. The differences were statistically significant at 60 minutes after starting CPB until the 6 hours post-operative. IL-18 plasma levels (S9 Fig) to all time points were substantially higher in non-survivors compared to survivors and differences became statistically significant 60 minutes after starting CPB until 6 hours post-operative. Plasm levels of IL-10 increased similarly in non-survivors and survivors after starting CPB, however, it remained significantly higher to all post-operative time points in non-survivors compared to survivors (S11Fig). 

Thus, we can conclude that patients who died within 30 days of surgery had stronger inflammatory response than those who survived.

The aim of this study was to compare, for the first time, the peri-operative cytokines profiles of patients undergoing cardiac surgery for IE with those of patients with non-infectious valve disease. Although we gathered information about the pre-operative neurological disorders and the timing of surgery in the CRF of the study, these were not the main objectives of this pilot study. 

We hope that this revised version now meets your expectations and thank you and all the reviewers for your efforts that improved this manuscript. 

References 

1. Vincent JL, de Mendonca A, Cantraine F, Moreno R, Takala J, Suter PM, et al. Use of the SOFA score to assess the incidence of organ dysfunction/failure in intensive care units: Results of a multicenter, prospective study. Crit Care Med. 1998;26(11):1793-800.

2. Ferreira FL, Bota DP, Bross A, Melot C, Vincent JL. Serial evaluation of the SOFA score to predict outcome in critically ill patients. Jama-J Am Med Assoc. 2001;286(14):1754-8.

3. Diab M, Guenther A, Sponholz C, Lehmann T, Faerber G, Matz A, et al. Pre-operative stroke and neurological disability do not independently affect short- and long-term mortality in infective endocarditis patients. Clinical research in cardiology : official journal of the German Cardiac Society. 2016;105(10):847-57.

4. Singer M, Deutschman CS, Seymour CW, Shankar-Hari M, Annane D, Bauer M, et al. The Third International Consensus Definitions for Sepsis and Septic Shock (Sepsis-3). JAMA : the journal of the American Medical Association. 2016;315(8):801-10.

---

## [Decision Letter · Decision Letter 1]

13 Jan 2020

Changes in Inflammatory and Vasoactive Mediator Profiles During Valvular Surgery With or Without Infective Endocarditis: A Case Control Pilot Study

PONE-D-19-17603R1

Dear Dr. Diab,

We are pleased to inform you that your manuscript has been judged scientifically suitable for publication and will be formally accepted for publication once it complies with all outstanding technical requirements.

With kind regards,

Andrea Ballotta

Academic Editor

PLOS ONE

Additional Editor Comments (optional):

Thank you for the revised version of your manuscript. It sounds suitable for publication

Reviewers' comments:

Reviewer's Responses to Questions

**Comments to the Author**

1. If the authors have adequately addressed your comments raised in a previous round of review and you feel that this manuscript is now acceptable for publication, you may indicate that here to bypass the “Comments to the Author” section, enter your conflict of interest statement in the “Confidential to Editor” section, and submit your "Accept" recommendation.

Reviewer #1: All comments have been addressed

Reviewer #2: All comments have been addressed

2. Is the manuscript technically sound, and do the data support the conclusions?

Reviewer #1: Yes

Reviewer #2: Partly

3. Has the statistical analysis been performed appropriately and rigorously? 

Reviewer #1: Yes

Reviewer #2: Yes

4. Have the authors made all data underlying the findings in their manuscript fully available?

Reviewer #1: Yes

Reviewer #2: Yes

5. Is the manuscript presented in an intelligible fashion and written in standard English?

Reviewer #1: Yes

Reviewer #2: Yes

6. Review Comments to the Author

Reviewer #1: (No Response)

Reviewer #2: (No Response)

7. PLOS authors have the option to publish the peer review history of their article (what does this mean?). If published, this will include your full peer review and any attached files.

Reviewer #1: No

Reviewer #2: No

---

## [Editor Report · Acceptance letter]

21 Jan 2020

PONE-D-19-17603R1 

Changes in Inflammatory and Vasoactive Mediator Profiles During Valvular Surgery With or Without Infective Endocarditis: A Case Control Pilot Study 

Dear Dr. Diab:

I am pleased to inform you that your manuscript has been deemed suitable for publication in PLOS ONE. Congratulations! Your manuscript is now with our production department. 

With kind regards,

on behalf of

Dr. Andrea Ballotta 

Academic Editor

PLOS ONE